# Functional Analyses of Four *CYP1A1* Missense Mutations Present in Patients with Atypical Femoral Fractures

**DOI:** 10.3390/ijms22147395

**Published:** 2021-07-09

**Authors:** Nerea Ugartondo, Núria Martínez-Gil, Mònica Esteve, Natàlia Garcia-Giralt, Neus Roca-Ayats, Diana Ovejero, Xavier Nogués, Adolfo Díez-Pérez, Raquel Rabionet, Daniel Grinberg, Susanna Balcells

**Affiliations:** 1Department of Genetics, Microbiology and Statistics, Faculty of Biology, Universitat de Barcelona, CIBERER, IBUB, IRSJD, 08028 Barcelona, Spain; nugartondo@ub.edu (N.U.); airun91@gmail.com (N.M.-G.); monicaesteve57@gmail.com (M.E.); neroca@clinic.cat (N.R.-A.); kelly.rabionet@ub.edu (R.R.); 2Musculoskeletal Research Group, IMIM (Hospital del Mar Medical Research Institute), Centro de Investigación Biomédica en Red en Fragilidad y Envejecimiento Saludable (CIBERFES), ISCIII, 08003 Barcelona, Spain; ngarcia@imim.es (N.G.-G.); dovejero@imim.es (D.O.); 85382@parcdesalutmar.cat (X.N.); ADiez@parcdesalutmar.cat (A.D.-P.)

**Keywords:** CYP1A1, osteoporosis, atypical femoral fractures, bisphosphonates

## Abstract

Osteoporosis is the most common metabolic bone disorder and nitrogen-containing bisphosphonates (BP) are a first line treatment for it. Yet, atypical femoral fractures (AFF), a rare adverse effect, may appear after prolonged BP administration. Given the low incidence of AFF, an underlying genetic cause that increases the susceptibility to these fractures is suspected. Previous studies uncovered rare *CYP1A1* mutations in osteoporosis patients who suffered AFF after long-term BP treatment. *CYP1A1* is involved in drug metabolism and steroid catabolism, making it an interesting candidate. However, a functional validation for the AFF-associated *CYP1A1* mutations was lacking. Here we tested the enzymatic activity of four such *CYP1A1* variants, by transfecting them into Saos-2 cells. We also tested the effect of commonly used BPs on the enzymatic activity of the CYP1A1 forms. We demonstrated that the p.Arg98Trp and p.Arg136His CYP1A1 variants have a significant negative effect on enzymatic activity. Moreover, all the BP treatments decreased CYP1A1 activity, although no specific interaction with CYP1A1 variants was found. Our results provide functional support to the hypothesis that an additive effect between CYP1A1 heterozygous mutations p.Arg98Trp and p.Arg136His, other rare mutations and long-term BP exposure might generate susceptibility to AFF.

## 1. Introduction

Osteoporosis, which leads to increased skeletal fragility, is the most common metabolic bone disorder and over recent decades nitrogen-containing bisphosphonates (BP) have been the first-line treatment to prevent osteoporotic fractures. BP treatment has been confirmed to reduce bone mineral density (BMD) decay and osteoporotic fractures through an inhibitory effect on bone resorption [1,2] and several clinical trials [3] and systematic reviews [4,5] have proved the overall safety of this treatment. However, a number of adverse effects have also been described associated with the prolonged use of these drugs [6], including atypical femoral fractures (AFF) [7]. AFFs are anatomically distinct from osteoporotic fractures; they are characterized by their location in the subtrochanteric or diaphyseal region and occur after minimal or no trauma [8,9]. Several diverse risk factors have been described for AFF, which may be considered a multifactorial and heterogeneous condition. These include anatomical traits (greater femoral bowing, loss of thigh muscle [10], and standing lower limb alignment [11]), pharmacological treatments (glucocorticoids [12] and proton pump inhibitors [13]), and diseases associated with defects in collagen and diabetes mellitus [12]. Moreover, a genetic susceptibility has been proposed since AFF is also associated with Asian ethnicity [14] and monogenic diseases such as hypophosphatasia (HPP), X-linked hypophosphatemia, pycnodysostosis, osteopetrosis, osteoporosis pseudoglioma syndrome (OPPG), osteogenesis imperfecta (OI), and X-linked osteoporosis [15]. The present study is focused on BP-related AFF and its possible interaction with a genetic background [16]. Since BP-related AFF have a very low incidence in the general population, it has been hypothesized that rare underlying genetic causes may increase the susceptibility to sustain them [8,9,15,16].

In search of such rare variants, we previously performed whole-exome sequencing (WES) to investigate the genetic background of three sisters affected with AFF and three additional unrelated AFF cases, all with a history of long BP treatment [16]. We detected a heterozygous mutation in *GGPS1* that showed a severe decrease in enzyme activity together with oligomerization defects [16]. Besides this functionally valid candidate, another gene, *CYP1A1*, was found mutated in heterozygosis in all three sisters and one unrelated patient [16]. Interestingly, this gene was also found mutated in two out of 17 AFF patients from a different study (11.8%) [17].

CYP1A1 belongs to the cytochrome P450 family, responsible for catalyzing the oxidative biotransformation of most drugs and other lipophilic xenobiotics, as well as endogenous steroids, cholesterol and fatty acids [18,19]. CYP1A1, in particular, is mainly expressed in extrahepatic tissue [18] and is considered to be the major CYP enzyme responsible for the 2-hydroxylation of 17-estradiol (E2) and estrone (E1) in extrahepatic tissues [19]. However, a variety of metabolites of different estrogenic activities can be formed by CYP1A1 hydroxylation [20,21]. Variants in CYP1A1 have been associated with alterations in estrogen metabolism and with differences in the 16-hydroxyestrone/2-hydroxyestrone metabolite balance [20,21]. Based on this relationship between estrogen metabolism and bone maintenance [21], Napoli et al. hypothesized that certain *CYP1A1* polymorphisms affecting enzymatic capacity, and therefore resulting in specific estrogen metabolic profiles, might be responsible for differences in BMD [22]. Due to the involvement of this protein family in drug metabolism, as well as its particular role in estrogen metabolism, CYP1A1 is an interesting AFF candidate for functional validation.

Interestingly, estrogen metabolism is located downstream of the mevalonate pathway where nitrogen-containing BPs exert their inhibitory effect. Nitrogen-containing BPs target and inhibit farnesyl pyrophosphate synthase (FFPPS) and geranylgeranyl diphosphate synthase (GGPPS), although to a lesser extent [1,2,23]. This interferes with the synthesis of the isoprenoid lipids farnesyl diphosphate and geranylgeranyl diphosphate, required for the post-translational prenylation of proteins necessary for cell survival [1,2,23,24]. The most commonly used BPs, alendronate, zoledronate and risedronate, have several differentiating characteristics [1,2,25]. Zoledronate has the strongest mineral binding affinity, which allows for the longest duration of action and the highest potency [25,26]. Moreover, zoledronate has been observed to inhibit osteoclastogenesis [27]. Subtoxic concentrations of zoledronate have also been seen to have an effect in osteoblasts [28].

In total, four rare variants of *CYP1A1* have been documented in BP-treated AFF patients: p.Arg98Trp (rs754416936) in three sisters, p.Ser216Cys (rs146622566) in an unrelated patient of the same study [16] and p.Arg136His (rs202201538) and p.Val409Ile (rs769134905) in two patients from the study by Peris et al. [17]. Here we present an in vitro assay in which two of them, p.Arg98Trp and p.Arg136His, display significantly reduced enzymatic activity. In addition, through testing the CYP1A1 enzymatic activity under several BP treatments, we observed a negative effect of BP exposure regardless of the variant.

## 2. Results and Discussion

The possibility to study three sisters with the same history of AFF after prolonged BP treatment against osteoporosis opened a door to find gene variants that could be related to their condition. CYP1A1 was presented as a possible candidate for BP-mediated AFF susceptibility after the finding of a rare variant in a WES study of these three AFF-affected sisters [16], and this candidacy was reinforced by the discovery of three additional rare variants in this gene in unrelated AFF patients [16,17]. In this study, we tested the effect of these four CYP1A1 variants on protein activity and their interaction with clinically used BPs.

CYP1A1 exerts its function in cholesterol metabolism downstream of the nitrogen-containing BP-mediated inhibition of the mevalonate pathway, and it is also responsible for drug and estrogen metabolism [18,19]. Imbalances in estrogen metabolism are a known cause for low BMD and some CYP1A1 variants have been observed to disturb this balance [20,21,22]. Moreover, a variant from another member of the CYP family, rs1934951 in CYP2C8, was associated with a trend to develop osteonecrosis of the jaw, another adverse effect associated with long BP exposure [29,30].

### 2.1. Location, Conservation and Pathogenicity Predictions of Four CYP1A1 Variants

Structurally, CYP1A1 is composed of twelve α-helices and three β-sheets [31]. The residues mutated in the patients included in the present study (p.Arg98Trp, p.Arg136His, p.Ser216Cys, and p.Val409Ile) are located in a turn (Arg98), in α-helices C and F (Arg136 and Ser216, respectively) and in the third β-sheet (Val409) [31] (Figure 1A). The active site of the protein is deeply buried within the structure, and none of these residues seem to directly take part in substrate binding, although they could still be responsible for enabling the substrate to reach the catalytic site (Figure 1A). The F helix is thought to be buried in the endoplasmic reticulum membrane and involved in enabling the access of hydrophobic ligands to the active site [32]. A few more structures have been obtained from crystals of CYP1A1 bound to different substrates. A recent crystallography study by Bart et al. suggested that CYP1A1 could have the flexibility to adapt to a wider range of substrates than previously thought [33]. The different studies give a more complete image of the substrate affinity and activity of the protein and report an adaptive flexibility provided specially by the F helix, in which the p.Ser216Cys variant is located [33].

The mutations found in AFF patients were located in moderately conserved positions of the protein. Among the observed vertebrate species, variant p.Val409Ile was also found in *Xenopus* and *Gallus* (Figure 1B). This same variant was also predicted as benign by most pathogenicity predictors, while the other three were considered pathogenic at least by SIFT (p.Ser216Cys), SIFT and REVEL (p.Arg136His), or all three pathogenicity prediction tools used (p.Arg98Trp) (Figure 1C).

### 2.2. Enzyme Activity Assays of Four CYP1A1 Variants

Upon transfection into Saos-2 cells, no effect on cell viability was observed among the four variants (Figure 2A). We also observed that mRNA expression of CYP1A1 was maintained at the same levels for the wild type and the four tested variants, up to three days after transfection (Appendix A). Therefore, any activity change observed in the mutant enzymes could not be attributable to an alteration at transcriptional level. The enzymatic activity of CYP1A1 p.Arg98Trp and p.Arg136His, tested by means of a luminescence assay in intact cells, was significantly decreased (*p* < 0.01) as compared to wild-type CYP1A1, reaching levels below 10% and around 20% of it, respectively. The other two variants showed no decrease (p.Ser216Cys) or a non-significant trend towards higher activity (p.Val409Ile; Figure 2B) when compared to the wild type. This is in agreement with the functional predictions by different algorithms (Figure 1C). Severe loss of function variants are uncommon findings for CYP1A1, as for most of the members of the CYP family [18], and it would be tempting to assume that rare CYP1A1 variants might be all pathogenic. In this line, our finding that two of the four examined variants did not show any compromised activity highlights the importance of functional studies. Moreover, the non-significant trend to an increased activity of variant p.Val409Ile is reminiscent of previous results for a nearby variant, p.Ile462Val, showing higher enzymatic activity [20]. It could be hypothesized that these two mutations (p.Val409Ile and p.Ile462Val) disturb a putative negative regulatory domain, not previously described. It is worth noting that mutations located closer to the C-terminus resulted in a lesser effect.

### 2.3. Effect of BP Treatment on Wild-Type CYP1A1 Enzyme Activity

We next tested the effect of BPs on CYP1A1 enzymatic activity. We selected three of the most prescribed nitrogen-containing BPs in the clinical setting, namely alendronate, zoledronate and risedronate. We selected two BP concentrations (1 μM and 5 μM) taking into account the normal intake for osteoporosis patients (2–4 mg/month intravenous zoledronate; 70 mg/week oral alendronate; 35 mg/week oral risedronate) [2], the low intestinal absorption (between 0.6% and 3% of an orally administered BP) [2], and that the plasma concentration of zoledronate after a 2–4 mg intake reaches 1 μM [34]. However, due to the high affinity of BPs to hydroxyapatite crystals, the concentration in bone tissue is expected to be higher. In this sense, Scheper et al. determined the concentration of zoledronate in bone in osteonecrosis patients to be between 0.4 μM and 4.6 μM [34].

Treatments did not significantly affect cell viability (Figure 3A), in agreement with previously observed results [28,34]. Wild-type CYP1A1 enzymatic activity was measured 48 h after BP treatment and, in all the cases, it was significantly decreased. This decrease reached 50% of the wild-type protein at 5 μM, and was lower but still significant at 1 μM (*p* < 0.001; Figure 3B).

### 2.4. Effect of BP Treatment on Enzyme Activity of Four CYP1A1 Variants

Since we had observed that both BP treatment and some of the CYP1A1 variants exerted a negative effect on CYP1A1 activity, we wondered whether a synergistic effect might occur when combining them. We used the same BP doses as before to test the four variants and observed a drop of activity in all mutants under all BP treatments (Figure 4). However, by normalizing each mutant activity to the wild-type activity under the same BP treatment, we did not observe any additional differences in the drop of each variant with respect to the wild type observed in the untreated condition and thus, we could discern that the BP effect size was independent of the mutation. Even if p.Arg98Trp and p.Arg136His showed a larger decrease by the combination of the BP effect and the mutation, the decrease was comparable with respect to the wild type of each condition. Therefore, the BP effect that we have described seems to act independently of the presence of the variants. Still, the additive effect of heterozygous mutations p.Arg98Trp and p.Arg136His found in the patients and the long-term BP exposure may result in a large decrease in CYP1A1 activity.

Even if the effects of BPs have been thoroughly investigated previously [23,25], to the best of our knowledge, this inhibitory effect of BPs on CYP1A1 activity is described here for the first time. Needless to say, it will be necessary to further study it in a more physiological context. 

### 2.5. AFF as an Oligogenic Phenotype; a Role for CYP1A1 Variants in Combination with BPs

The possibility to study three sisters with a similar clinical history has provided us an opportunity to unveil the genetic factors associated with AFF. We previously reported a mutation in a gene (*GGPS1*) shared by the sisters, which resulted in a loss of enzymatic activity [16]. In the current study, we confirmed that the *CYP1A1* variant p.Arg98Trp, also shared by the sisters, resulted in a decrease in CYP1A1 enzymatic activity. We cannot rule out an oligogenic effect of both mutations in AFF susceptibility. Moreover, we have proved that the CYP1A1 variant p.Arg136His, identified in an unrelated case of AFF, also decreased the enzyme functionality, suggesting that CYP1A1 could have a role in the AFF susceptibility, with two mutations with a deleterious effect found in two unrelated families. In the case of variants p.Ser216Cys and p.Val409Ile, without an effect on CYP1A1 enzymatic activity, we hypothesize that other variants in other genes may have a role in the development of AFF. It is worth noting that there were no evident clinical differences among the AFF patients carrying these four variants, all having osteoporosis and hypercholesterolemia and no hypophosphatasia. Patients carrying variants p.Ser216Cys and p.Arg136His were diabetic. Further information of the patients can be found in Peris et al. [17] and Roca-Ayats et al. [16].

In addition, we observed that BP treatments negatively affected all CYP1A1 variants, including the wild type. Therefore, both BP treatment and the variants p.Arg98Trp and p.Arg136His decreased CYP1A1 activity in the respective carriers. All in all, a hypothesis might arise that a combination of the negative effect of BPs on both the wild-type and mutant copy of CYP1A1, along with the already decreased activity of p.Arg98Trp or p.Arg136His variants, results in an overall lower enzymatic activity of CYP1A1, below a hypothetical pathogenic threshold. 

The present study showed that two of the studied CYP1A1 mutations have an effect on the enzyme activity and, thus, might contribute to an enhanced susceptibility to AFF. However, AFF etiology is multifactorial, involving an as yet incompletely characterized mix of genetic, anatomical, and environmental factors, which calls for further research. Only after the analysis of more cases to obtain a more detailed and focused picture of all the AFF players will we be able to design proper prevention measures, which may involve both genetic testing and tailored adjustment of osteoporosis treatment schedules.

## 3. Materials and Methods

### 3.1. In Silico Prediction of Protein Structure and Pathogenicity

We used ClustalW and EsPript [35] to compare the amino acid sequences of several vertebrate CYP1A1 proteins and to locate the variants in the structure of the protein (PDB structure provided by Walsh et al. (4I8V) [31]. The predicted pathogenicity of the variants was calculated with the predictors of the SNP effect on functioning of SIFT (http://sift.bii.a-star.edu.sg/, accessed on 8 January 2021), PolyPhen (http://genetics.bwh.harvard.edu/pph2/, accessed on 8 January 2021), and REVEL (accessed on 8 January 2021) [36].

### 3.2. Site-Directed Mutagenesis

The mammalian expression vector pcDNA3.1(+) containing the human CYP1A1 wild-type cDNA (NM_000499) was purchased from GenScript (Piscataway, NJ, USA). The desired mutations were inserted using a QuickChange Lightning Site-Directed Mutagenesis Kit (Agilent, Santa Clara, CA, USA) following the commercial protocol and using the primers shown in Appendix A.

### 3.3. Cell Culture and Transfection

Saos-2 cells were chosen due to the low endogenous expression of CYP1A1 and their osteoblast-like characteristics. Cells were cultured in DMEM (Sigma-Aldrich, St. Louis, MO, USA) with 10% FBS and 1% penicillin/streptomycin, in 5% CO_2_ wet conditions at 37 °C. Cells were seeded in 6-well plates (300,000 cells/well) for RNA extraction and in 96-well plates (10,000 cells/well) for cell viability and enzymatic assays. Twenty-four hours later, they were transfected (Fugene HD transfection Reagent, Promega, Madison, WI, USA) with either the empty vector pcDNA3.1+ (Invitrogen, Waltham, MA, USA), or the vector containing the wild-type CYP1A1, or any of the four mutated forms. Transfection was performed by mixing 2 µg of DNA and 8 µL of Fugene per well in the case of 6-well plates and 100 ng of DNA and 0.4 µL of Fugene per well in the case of 96-well plates.

### 3.4. RT-qPCR

RNA extraction was performed 48 h post-transfection using the High Pure RNA Isolation Kit (Roche, Basel, Switzerland) following the manufacturer’s recommendations. 2 µg of RNA were used for retro-transcription with the High-Capacity cDNA Reverse Transcription Kit (Applied Biosystems, Foster City, CA, USA), followed by RT-qPCR with LightCycler® 480 Probes Master (Roche, Basel, Switzerland) in a LightCycler480 (Roche, Basel, Switzerland) following the commercial protocol. A TaqMan assay (Hs01054796_g1) for CYP1A1 expression quantification (Appendix A) was used and GADPH primers combined with UPL probe (#63; Roche) were used as endogenous control.

### 3.5. Bisphosphonate Treatment and Enzymatic and Viability Assays

Alendronate, zoledronate, and risedronate (Sigma-Aldrich, St. Louis, MO, USA) were resuspended and diluted at 1 μM and 5 μM in DMEM 10% FBS and 1% penicillin/streptomycin. Cells were seeded in 96-well plates, transfected 24 h later and media was replaced with BP-containing media 24 h post-transfection. Viability and enzymatic assays were performed 48 h after BP exposure. Enzymatic activity was measured by the conversion of a CYP1A1 proluciferin substrate into a luciferin product detectable by luminescence using a P450-Glo Assay (Promega, Madison, WI, USA). The non-lytic assay was performed directly in the cultured cells following the commercial protocol. Viability assays were performed by rinsing the wells in PBS and adding 100 µL/well of MTT (Sigma-Aldrich, St. Louis, MO, USA) diluted 1:10 in DMEM. Cells were incubated for 3 h at 37 °C. Then, the media was aspired and 100 µL/well of DMSO (Merck-Millipore, Burlington, MA, USA) were added. Plates were shaken at 150 rpm for 5 min and media was transferred to a transparent plate for its quantification in a Modulus Microplate (Thermo Fisher Scientific, Waltham, MA, USA) at 560 nm.

### 3.6. Statistical Analysis

The effects of variants and BP treatments on cell viability were analyzed with ANOVA and Tukey’s HSD tests. The enzymatic activity of CYP1A1 variants was evaluated by the non-parametric Kruskal–Wallis test and pairwise Wilcox test with Bonferroni correction. Activities were expressed as a percentage with respect to that of the wild-type protein. Regarding the effect of different BPs on wild-type CYP1A1 activity, changes in activity were analyzed by ANOVA and Tukey’s HSD tests. The enzymatic activities of variants were normalized with the enzymatic activity of the wild-type well from each treatment and analyzed again by ANOVA and Tukey’s HSD tests. All statistical analyses were executed in R 3.6.3 and graphic representations on GraphPad 6.01. All experiments were performed thrice including three replicas each and p values < 0.05 were considered significant.

## 4. Conclusions

Our results provide functional support to the hypothesis that an additive effect between CYP1A1 heterozygous mutations p.Arg98Trp or p.Arg136His and long-term BP exposure may generate susceptibility to AFF. Combination of rare mutations and exposure to BP might be at the origin of some cases of AFF.

## Figures and Tables

**Figure 1 ijms-22-07395-f001:**
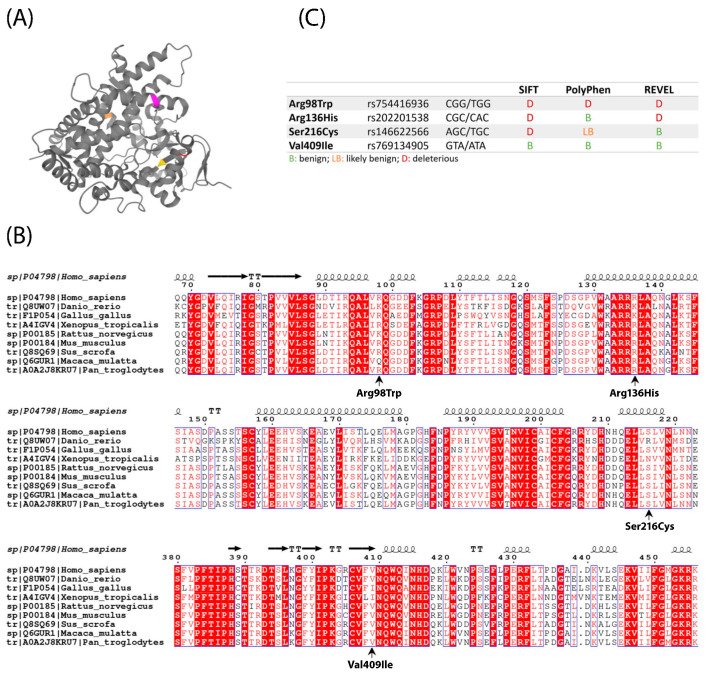
(**A**) PDB structure (4I8V) of human CYP1A1 with the mutated residues highlighted in different colors: Arg98 (red), Arg136 (pink), Ser216 (orange), and Val409 (yellow). (**B**) Partial alignment of CYP1A1 protein sequences from several vertebrate species, with the four AFF-associated mutations indicated under it. Species included are human (*Homo sapiens*), macaque (*Macaca mulata*), chimpanzee (*Pan troglodytes*), mouse (*Mus musculus*), rat (*Rattus norvegicus*), pig (*Sus scrofa*), zebrafish (*Danio rerio*), Western clawed toad (*Xenopus tropicalis*) and chicken (*Gallus gallus*). (**C**) Predictions of pathogenicity of the four AFF-associated CYP1A1 variants according to SIFT, Poly-Phen and Revel.

**Figure 2 ijms-22-07395-f002:**
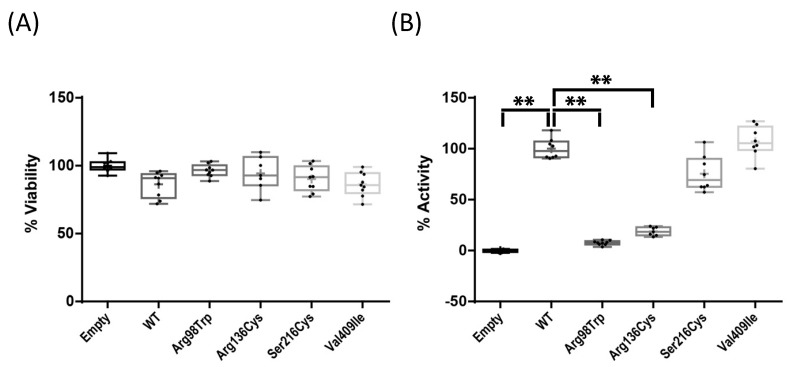
(**A**) Viability of cells transfected with the variants normalized to that of cells transfected with the empty vector, arbitrarily set at 100%. (**B**) Enzymatic activity of the CYP1A1 protein WT and with the variants Arg98Trp, Arg136His, Ser216Cys, and Val409Ile transfected in Saos-2 cells, normalized to the wild-type activity, arbitrarily set at 100% (** *p* < 0.01). All experiments were performed thrice including three replicas each and *p* values < 0.05 were considered significant.

**Figure 3 ijms-22-07395-f003:**
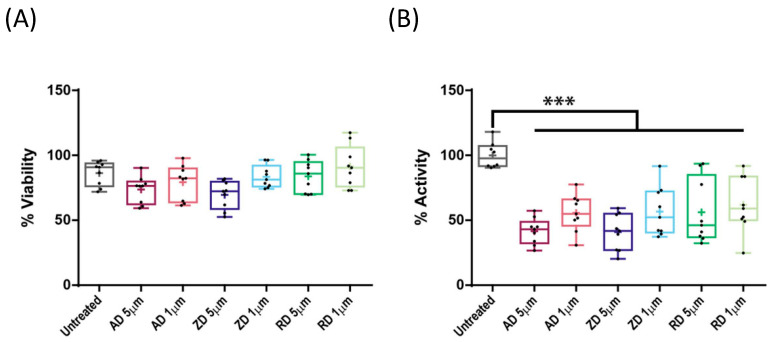
(**A**) Effects of alendronate (AD), zoledronate (ZA), and risedronate (RD) at the indicated concentrations on cell viability. (**B**) Effects of AD, ZD, and RD treatments, at the indicated concentrations, on wild-type CYP1A1 enzymatic activity (*** *p* < 0.001). Activities are expressed as percentages of the untreated wild type. All experiments were performed thrice including three replicas each and *p* values < 0.05 were considered significant.

**Figure 4 ijms-22-07395-f004:**
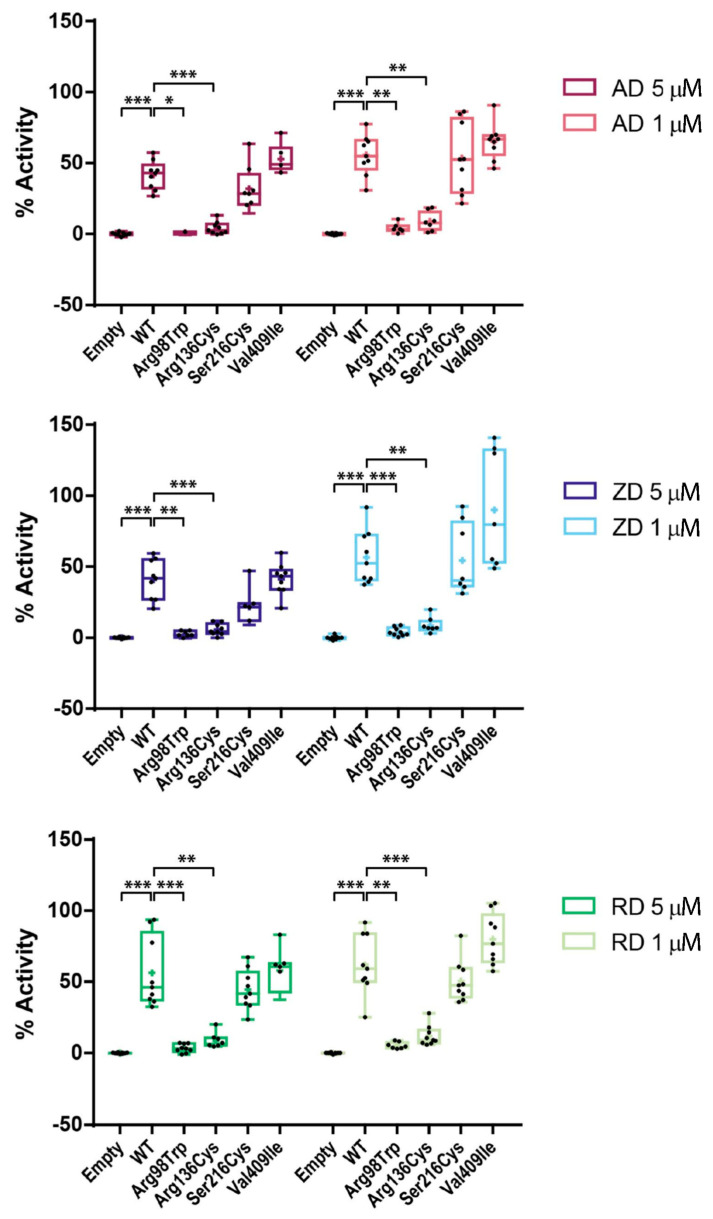
Effects of alendronate (AD), zoledronate (ZA), and risedronate (RD) at the indicated concentrations on the enzymatic activity of wild-type CYP1A1 protein and CYP1A1 variants Arg98Trp, Arg136His, Ser216Cys, and Val409Ile (* *p* < 0.05; ** *p* < 0.01; *** *p* < 0.001). Activities are expressed as percentages of the untreated wild type. All experiments were performed thrice including three replicas each and *p* values < 0.05 were considered significant.

## Data Availability

Not applicable.

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
