# Peer review of "Functional Analyses of Four CYP1A1 Missense Mutations Present in Patients with Atypical Femoral Fractures"

_ijms, 2021, doi:10.3390/ijms22147395_

Round 1
Reviewer 1 Report
Ugartondo et al. have previously shown rare CYP1A1 mutations in osteoporosis patients who suffered AFF after long-term BP treatment. They have now tested the enzymatic activity of four such rare CYP1A1 variants, by transfecting them into Saos-2 cells. They demonstrated that two of the four CYP1A1 variants, p.Arg98Trp and p.Arg136His, have a significant negative effect on enzymatic activity. Moreover, three commonly used bisphosphonates also decreased CYP1A1 activity, although no specific interaction with any of the CYP1A1 variants was identified. The authors conclude that an additive effect between the CYP1A1 heterozygous mutations, p.Arg98Trp and p.Arg136His, and long-term bisphosphonate exposure may generate susceptibility to AFF.
This reviewer has the following comments/questions:
- Abstract: "strong" could be deleted describing the negative effect of the variants on enzyme activity.
- Section 2.4, line 178: after normalising mutant activity to wild type, was there a difference of bisphosphonate therapy on enzyme activity for either p.Arg98Trp and p.Arg136His? This needs to be more clearly stated in the paper.
- Line 189; "drastic" could be replaced by 'large"
- In the oligogenic phenotype of AFF claimed by the authors, which mutations are more important, those of GGPS1 or CYP1A1? How could this be determined?
- Could the oligogenic phenotype be included in the abstract conclusion to match the paper's conclusion?
Author Response
Point by point answer to reviewer’s comments
Reviewer 1
Comments and Suggestions for Authors
Ugartondo et al. have previously shown rare CYP1A1 mutations in osteoporosis patients who suffered AFF after long-term BP treatment. They have now tested the enzymatic activity of four such rare CYP1A1 variants, by transfecting them into Saos-2 cells. They demonstrated that two of the four CYP1A1 variants, p.Arg98Trp and p.Arg136His, have a significant negative effect on enzymatic activity. Moreover, three commonly used bisphosphonates also decreased CYP1A1 activity, although no specific interaction with any of the CYP1A1 variants was identified. The authors conclude that an additive effect between the CYP1A1 heterozygous mutations, p.Arg98Trp and p.Arg136His, and long-term bisphosphonate exposure may generate susceptibility to AFF.
This reviewer has the following comments/questions:
- Abstract: "strong" could be deleted describing the negative effect of the variants on enzyme activity.
Taking the advice in consideration, we have deleted this word from the abstract.
- Section 2.4, line 178: after normalising mutant activity to wild type, was there a difference of bisphosphonate therapy on enzyme activity for either p.Arg98Trp and p.Arg136His? This needs to be more clearly stated in the paper.
Indeed, none of the mutations, including pArg98Trp and p.Arg136His, shows a difference in response to bisphosphonate therapy besides the already observed decrease. To clarify any concerns in the matter, we have included the following sentences in Results and discussion (line 190): “However, by normalizing each mutant activity to the wildtype activity under the same BP treatment, we did not observe any additional differences in the drop of each variant with respect of the wildtype observed in the untreated condition and thus, we could discern that the BP effect size was independent of the mutation. Even if p.Arg98Trp and p.Arg136His showed a larger decrease by the combination of the BP effect and the mutation, the decrease was comparable with respect to the wild type of each condition.”
- Line 189; "drastic" could be replaced by 'large"
Taking the advice in consideration, we have replaced this word from Results and discussion.
- In the oligogenic phenotype of AFF claimed by the authors, which mutations are more important, those of GGPS1 or CYP1A1? How could this be determined?
This is a good question, which would require the generation and characterization of single and double mutant animal models, which is beyond the scope of this paper but which may be approached in the future.
- Could the oligogenic phenotype be included in the abstract conclusion to match the paper's conclusion?
We agree with the reviewer and we have included this conclusion in the abstract (line 23): “Our results provide functional support to the hypothesis that an additive effect between CYP1A1 heterozygous mutations p.Arg98Trp and p.Arg136His, other rare mutations and long-term BP exposure might generate susceptibility to AFF.”

Reviewer 2 Report
Summary:
This manuscript reported a study about the functional role of four CYP1A1 missense mutations associated with atypical femoral fractures. This study assessed the enzymatic activity of four such CYP1A1 variants, by transfecting them into Saos-2 cells. They also tested the effect of commonly used BPs on the enzymatic activity of the CYP1A1 forms. The results showed that the p.Arg98Trp and p.Arg136His CYP1A1 variants have a significant and strong negative effect on enzymatic activity. In addition, all the BP treatments decreased CYP1A1 activity, although no specific interaction with CYP1A1 variants was found. The authors demonstrated functional support to the hypothesis that an additive effect between CYP1A1 heterozygous mutations p.Arg98Trp and p.Arg136His and long-term BP exposure generates susceptibility to AFF. The authors concluded that their results provided functional support to the hypothesis that an additive effect between CYP1A1 heterozygous mutations p.Arg98Trp or p.Arg136His and long-term BP exposure generates susceptibility to AFF. Combination of rare mutations and exposure to BP might be at the origin of some cases of AFF.
Comments:
This manuscript reported a study on the possible genetic mutations of CYP1A1 in the AFF oligogenic phenotype. They showed that both BP treatment and the variants p.Arg98Trp and p.Arg136His decreased CYP1A1 activity in the respective carriers. This study demonstrated a role for CYP1A1 variants in combination with BPs. This study provided some evidence of genetic variants of CYP1A1 and the potential bone metabolism disturbance. However, some issues need to be addressed before reaching a solid conclusion about the relationship between CYP1A1 mutation genotype anf their roles in the AFF oligogenic phenotype.
- As mentioned in Lines 76-82 and Lines 133-137, there are 4 rare variants of CYP1A1 have been documented in BP-treated AFF patients: p.Arg98Trp (rs754416936) in three sisters, p.Ser216Cys (rs146622566) in an unrelated patient of the same study [10] and p.Arg136His (rs202201538) and p.Val409Ile (rs769134905) in two patients from the study by Peris et al. [12]. However, this vitro assay showed p.Arg98Trp and p.Arg136His that displayed significantly reduced enzymatic activity. In addition, through testing the CYP1A1 enzymatic activity under several BP treatments, we observed a negative effect of BP exposure regardless of the variant.” “The other two variants showed no decrease (p.Ser216Cys) or a non-significant trend towards higher activity (p.Val409Ile; Figure 2B) when compared to the wild type.” Why the other two variants did not show similar effects? How much of the decreased activity be expected to be related to occurrence of AFFs?
- As mentioned in Lines 57-64, “Variants in CYP1A1 have been associated with alterations in oestrogen metabolism and with differences in the 16-hydroxyestrone/2-hydroxyestrone metabolite balance [15,16]. Based on this relationship between oestrogen metabolism and bone maintenance [16], Napoli et al. hypothesized that certain CYP1A1 polymorphisms affecting enzymatic capacity, and therefore resulting in specific oestrogen metabolic profiles, might be responsible for differences in BMD [17].” Those patients with CYP1A1 polymorphisms may present bone metabolism disorders. As you know, decrease of BMD did not mean the occurrence of AFFs. The authors may need to present the clinical profiles of these 4 patients. In addition, the etiology of AFFs is multi-factorial. Therefore this study may need to check the other risk factors before making a conclusion in this study.
- This study should show the incidence of the rare CYP1A1 mutations in patients with AFF, especially for those with bisphosphonate treatment. However, AFFs are not so rare after longer term BP treatment. On of the current concept is that BP can reduce much bone remodeling activities. The repair potential of such frozen bone was low and the initial microcrack may progress to AFFs finally. Is it possible that rare CYP1A1 mutations associated with lower potential of bone repair, especially for those patients under long-term bisphosphonate treatment.
- As mentioned in Lines 57-64, “certain CYP1A1 polymorphisms affecting enzymatic capacity, and therefore resulting in specific oestrogen metabolic profiles, might be responsible for differences in BMD [17]“ Did the authors find disturbance of estrogen metabolic profiles of these patients?
- Did the CYP1A1 polymorphisms affect bone cells directly, e.g., osteoblasts or osteoclasts?
- Some typos need correction.
Author Response
Point by point answer to reviewers’ comments
Reviewer 2
Comments and Suggestions for Authors
Summary:
This manuscript reported a study about the functional role of four CYP1A1 missense mutations associated with atypical femoral fractures. This study assessed the enzymatic activity of four such CYP1A1 variants, by transfecting them into Saos-2 cells. They also tested the effect of commonly used BPs on the enzymatic activity of the CYP1A1 forms. The results showed that the p.Arg98Trp and p.Arg136His CYP1A1 variants have a significant and strong negative effect on enzymatic activity. In addition, all the BP treatments decreased CYP1A1 activity, although no specific interaction with CYP1A1 variants was found. The authors demonstrated functional support to the hypothesis that an additive effect between CYP1A1 heterozygous mutations p.Arg98Trp and p.Arg136His and long-term BP exposure generates susceptibility to AFF. The authors concluded that their results provided functional support to the hypothesis that an additive effect between CYP1A1 heterozygous mutations p.Arg98Trp or p.Arg136His and long-term BP exposure generates susceptibility to AFF. Combination of rare mutations and exposure to BP might be at the origin of some cases of AFF.
Comments:
This manuscript reported a study on the possible genetic mutations of CYP1A1 in the AFF oligogenic phenotype. They showed that both BP treatment and the variants p.Arg98Trp and p.Arg136His decreased CYP1A1 activity in the respective carriers. This study demonstrated a role for CYP1A1 variants in combination with BPs. This study provided some evidence of genetic variants of CYP1A1 and the potential bone metabolism disturbance. However, some issues need to be addressed before reaching a solid conclusion about the relationship between CYP1A1 mutation genotype and their roles in the AFF oligogenic phenotype.
- As mentioned in Lines 76-82 and Lines 133-137, there are 4 rare variants of CYP1A1 have been documented in BP-treated AFF patients: p.Arg98Trp (rs754416936) in three sisters, p.Ser216Cys (rs146622566) in an unrelated patient of the same study [10] and p.Arg136His (rs202201538) and p.Val409Ile (rs769134905) in two patients from the study by Peris et al. [12]. However, this vitro assay showed p.Arg98Trp and p.Arg136His that displayed significantly reduced enzymatic activity. In addition, through testing the CYP1A1 enzymatic activity under several BP treatments, we observed a negative effect of BP exposure regardless of the variant.” “The other two variants showed no decrease (p.Ser216Cys) or a non-significant trend towards higher activity (p.Val409Ile; Figure 2B) when compared to the wild type.” Why the other two variants did not show similar effects? How much of the decreased activity be expected to be related to occurrence of AFFs?
This is an interesting point. Different mutations along the gene may have different effects in the development of AFF, ranging from an important contribution to no effect at all. In silico predictors allow us to define the more likely pathogenic among them but functional assays, as those performed in this study, are necessary to confirm the prediction. Under our hypothesis that AFF has an oligogenic and also a genetic heterogeneous origin, rare variants form the same gene may or may not have an effect on AFF. In the two AFF cases with non-pathogenic CYP1A1 variants, we hypothesize that other variants in other genes may be at play.
Thanks to the reviewer we have included these thoughts in the discussion in line 231: “In the case of variants p.Ser216Cys and p.Val409Ile, without an effect in CYP1A1 enzymatic activity, we hypothesize that other variants in other genes may have a role in the development of AFF.”
- As mentioned in Lines 57-64, “Variants in CYP1A1 have been associated with alterations in oestrogen metabolism and with differences in the 16-hydroxyestrone/2-hydroxyestrone metabolite balance [15,16]. Based on this relationship between oestrogen metabolism and bone maintenance [16], Napoli et al. hypothesized that certain CYP1A1 polymorphisms affecting enzymatic capacity, and therefore resulting in specific oestrogen metabolic profiles, might be responsible for differences in BMD [17].” Those patients with CYP1A1 polymorphisms may present bone metabolism disorders. As you know, decrease of BMD did not mean the occurrence of AFFs. The authors may need to present the clinical profiles of these 4 patients. In addition, the etiology of AFFs is multi-factorial. Therefore this study may need to check the other risk factors before making a conclusion in this study.
According to the reviewer’s suggestion, we have added the following information in line 234: “It is worth noting that there were no evident clinical differences among the AFF patients carrying these four variants, all having osteoporosis and hypercholesterolemia and no hypophosphatasia. Patients carrying variants p.Ser216Cys and p.Arg136His were diabetic. Further information of the patients can be found in Peris et al. [12] and Roca-Ayats et al. [10].”
- This study should show the incidence of the rare CYP1A1 mutations in patients with AFF, especially for those with bisphosphonate treatment. However, AFFs are not so rare after longer term BP treatment. One of the current concept is that BP can reduce much bone remodeling activities. The repair potential of such frozen bone was low and the initial microcrack may progress to AFFs finally. Is it possible that rare CYP1A1 mutations are associated with lower potential of bone repair, especially for those patients under long-term bisphosphonate treatment?
Due to the recent association of these mutations with AFF and the small sample sizes, we do not currently have an accurate incidence of these mutations among AFF patients. We expect that this situation will change after the publication of this study if sequencing of CYP1A1 is undertaken in other AFF patient studies. Nevertheless, we have added the incidence as for the paper of Peris et al. in the introduction in line 56: ”Interestingly, this gene was also found mutated in two out of 17 AFF patients from a different study (11,8%) [12].”
- As mentioned in Lines 57-64, “certain CYP1A1 polymorphisms affecting enzymatic capacity, and therefore resulting in specific oestrogen metabolic profiles, might be responsible for differences in BMD [17]“ Did the authors find disturbance of estrogen metabolic profiles of these patients?
Although this information would be very interesting, these data are not available.
- Did the CYP1A1 polymorphisms affect bone cells directly, e.g., osteoblasts or osteoclasts?
This is a very interesting point that we may approach in the future in bone specific cells by CRISPR.
- Some typos need correction.
We have checked the writing of the document.

Round 2
Reviewer 2 Report
This revised manuscript did not provide adequate evidence for supporting the hypothesis raised in the study. More analysis of multiple clinical risk factors in more cases are needed before reaching a solid conclusion.
Author Response
This revised manuscript did not provide adequate evidence for supporting the hypothesis raised in the study. More analysis of multiple clinical risk factors in more cases are needed before reaching a solid conclusion.
We absolutely agree with reviewer 2 in that more research is needed to comprehensively understand the etiology of AFF.
To address the concerns raised by reviewer 2 regarding the MS we have modified it in three ways:
- We have included a paragraph in the Introduction (lines 39-53) in which AFF is explained as a clinical entity with multifactorial and heterogeneous aetiology, and in which the known AFF risk factors are detailed and referenced. We hope that now this paragraph serves as a wide framework in which our particular research is properly dimensioned.
- In the results and discussion section (lines 337-338 and 353-359), we conclude that the loss of enzyme activity of two CYP1A1 variants together with a reduction of CYP1A1 activity exerted by bisphosphonates (that we have observed in vitro) may participate -together with other factors- to confer susceptibility to AFF. And we state that further research in further AFF cases will be necessary to clarify all the AFF risk factors.
We hope that this wording is clearer in that we are not pretending to attribute AFF causality to the variants but instead, we are suggesting that two particular CYP1A1 variants may have contributed to AFF because they are proven loss-of-function variants.
From the geneticist point of view, a variant whose effect is not significantly different from the wild type will not possibly be involved in disease causality. Therefore, it is essential to test the functionality of genetic variants found in patients and to discern if they are loss-of-function or not. And this is what we have done. We obtained an interesting result because of four tested variants, two were loss-of-function and the other two were not. The former may contribute to AFF susceptibility while the latter may not. We do hope that this is now clearer to the reader.
- We have modified the title to remove the “association between CYP1A1 variants and AFF” and instead to describe that the variants studied were “present in AFF patients”.
